# A Method for Extracting High-Resolution Building Height Information in Rural Areas Using GF-7 Data

**DOI:** 10.3390/s24186076

**Published:** 2024-09-20

**Authors:** Mingbo Liu, Ping Wang, Kailong Hu, Changjun Gu, Shengyue Jin, Lu Chen

**Affiliations:** National Disaster Reduction Center of China, Ministry of Emergency Management of the People’s Republic of China, Beijing 100124, China; wangping@ndrcc.org.cn (P.W.); hukailong@ndrcc.org.cn (K.H.); guchangjun@ndrcc.org.cn (C.G.); jinshengyue@ndrcc.org.cn (S.J.); chenlu@ndrcc.org.cn (L.C.)

**Keywords:** GF-7 image, building height, rural areas, deep learning

## Abstract

Building height is important information in disaster management and damage assessment. It is also a key parameter in studies such as population modeling and urbanization. Relatively few studies have been conducted on extracting building height in rural areas using imagery from China’s Gaofen-7 satellite (GF-7). In this study, we developed a method combining photogrammetry and deep learning to extract building height using GF-7 data in the rural area of Pingquan in northern China. The deep learning model DELaMa was proposed for digital surface model (DSM) editing based on the Large Mask Inpainting (LaMa) architecture. It not only preserves topographic details but also reasonably predicts the topography inside the building mask. The percentile value of the normalized digital surface model (nDSM) in the building footprint was taken as the building height. The extracted building heights in the study area are highly consistent with the reference building heights measured from the ICESat-2 LiDAR point cloud, with an R^2^ of 0.83, an MAE of 1.81 m and an RMSE of 2.13 m for all validation buildings. Overall, the proposed method in this paper helps to promote the use of satellite data in large-scale building height surveys, especially in rural areas.

## 1. Introduction

Building height information plays an important role in disaster management [1,2,3,4] and damage assessment [5]. It is also widely used in studies such as population modeling and urbanization [6]. Remote sensing technology provides an effective tool for large-scale building height surveys. Light detection and ranging (LiDAR), synthetic aperture radar (SAR), and high-resolution optical images are used to extract building height [7]. LiDAR measures building height using waveform decomposition or point cloud data. Nevertheless, obtaining building height information over large areas using LiDAR is a challenging task due to the high cost of airborne LiDAR and the incomplete coverage of spaceborne LiDAR [8,9,10,11,12,13]. The building height can also be extracted from the SAR radar signal. However, the complex signals resulting from different scattering mechanisms increase the uncertainty of the building height extraction [14,15,16]. In contrast, sub-meter resolution optical imagery has shown great potential in the extraction of building height information over large areas.

For a single optical image, building height can be derived from the geometric relationship between the sun, the sensor, and the length of the building’s shadow, but the accuracy is limited for low-rise buildings or when the shadow is blocked [7,17,18,19]. Another method is to extract building height information from multi-view optical imagery based on a disparity map [20,21,22,23,24]. The principle is to calculate the difference between the digital surface model (DSM) and the digital terrain model (DTM) or the ground elevation to obtain the normalized digital surface model (nDSM), which contains the building height information. Liu et al. [22] extracted building height in Wuhan from ZY-3 stereo images by generating a DSM with the semi-global matching (SGM) algorithm and estimating the ground elevation with the top-hat morphological filtering. Similarly, Wang et al. [25] extracted a DSM in Beijing using GF-7 stereo images and SGM. The cloth simulation filter (CSF) was employed to separate ground points. Zhang et al. [26] extracted DSM in Yingde and Xi’an using GF-7 stereo images and roof contour-constrained stereo matching. The bottom elevation was estimated according to the elevation histogram in the building buffer. Chen et al. [21,27] used GF-7 images and a StereoNet-based deep learning method to generate DSMs in eight Chinese cities and used the minimum value of the DSM in the building buffer to represent the ground elevation. Cui et al. [28] used GF-7 images and contour matching to extract roof elevation in Yingde, Guangzhou, and Xi’an and used classification results from multispectral imagery and a ground filtering algorithm to obtain the ground elevation.

In summary, previous studies have demonstrated the feasibility of high-resolution optical stereo images for large-scale building height extraction. Especially in recent years, with the increase of available data, methods for building height extraction using optical stereo images have been rapidly tested and optimized. However, we found that most of these studies were carried out in large cities, and few studies have been done on extracting building height in rural areas. Building height information in rural areas is important. For example, some natural disasters, such as landslides and torrents, are more likely to occur in rural areas, and building height information is needed to assess the damage caused by these disasters [5]. It is also important from a research point of view because buildings in rural areas are very different from those in urban areas in several ways: (1) more industrial and agricultural buildings, (2) more low-rise and small buildings with more pitched roofs, and (3) uneven spatial and clustered distribution. In addition, methods using optical stereo images derive building height from the difference between the DSM and the DTM. It is common practice to use filtering and interpolation to obtain a flat and smooth DTM, or to use the minimum elevation within the building buffer as the ground elevation. In urban areas, this is feasible and efficient. However, in rural areas, especially mountainous areas, new methods are needed to deal with the effects of dense, low-rise buildings and topography. The objectives of this study are as follows:Evaluate the capability and limitations of GF-7 images for building height extraction in rural areas.Develop a method for automatic DTM generation based on deep learning architecture.

The remainder of this paper is arranged as follows. Section 2 introduces the study area and data. Section 3 describes the workflow and methodology. The results are presented in Section 4 and discussed in Section 5. Section 6 summarizes the paper.

## 2. Study Area and Data

### 2.1. Study Area

The study area is located in Pingquan, Hebei Province, China (Figure 1b). It has an area of 562 km^2^ and elevations ranging from 520 m to 1181 m above sea level (Figure 1a), with longitude ranging from 118°32′43″ E to 118°52′20″ E and latitude ranging from 41°2′27″ N to 41°17′49″ N. The main land cover types in the study area are forest, shrub, farmland, buildings, and water. The main building types are bungalow, workshop, multistory, and greenhouse (Figure 1c). The diversity of topography and industry makes the study area a representative sample of rural areas. In the northwestern part of the study area, there are villages in the mountains where the predominant building style is the bungalow. The northeastern part of the study area, which is a relatively flat and open area where the rivers meet, is much more densely populated, with a large number of greenhouses and workshops. The southeastern part of the study area is close to the center of Pingquan City and has more large factories and office buildings.

### 2.2. GF-7 Data

The GF-7 satellite was launched in November 2019 and is mainly used for natural resource monitoring and land surveying. The satellite runs on a 500 km sun-synchronous orbit. It is equipped with a two-line array scanner with a forward inclination of 26°, a backward inclination of −5°, and a base-height ratio of about 0.6. The forward panchromatic imagery, backward panchromatic imagery, and four-band backward multispectral imagery have spatial resolutions of 0.8 m, 0.65 m, and 2.6 m, respectively, with a ground swath wider than 20 km [29,30]. The planar residual of stereo mapping is about 1.4 m, and the vertical residual is about 0.86 m when 5 or 9 control points are used, as verified by Li et al. [31]. The GF-7 data used in this study were obtained on 3 April 2020 and downloaded from the China Centre for Resources Satellite Data and Application (CRESDA; https://data.cresda.cn/#/home, accessed on 2 September 2024).

### 2.3. ICESat-2/ATLAS

The ICESat-2 satellite was launched in September 2018. Equipped with the Advanced Topographic Laser Altimeter System (ATLAS), it uses laser pulses to measure the height of the Earth’s surface. Three of the six beams emitted by the laser are strong beams, and the other three are weak beams with an energy ratio between them of approximately 4:1. The transmitted laser pulse measures the surface elevation with an accuracy of 0.1 m [32,33]. We acquired the ICESat-2 L2A Global Geolocated Photon Data (ATL03) and the L3A Land and Vegetation Height Data (ATL08) over the study area from 1 January 2020 to 1 May 2023. The date interval was chosen based on the principle of ensuring sufficient data for validation. We also referenced other high-resolution remote sensing data to ensure that the landscape did not change significantly during this period. The ICESat-2 ATL03 and ATL08 products were downloaded from the NASA Earthdata website (https://search.earthdata.nasa.gov/search, accessed on 3 September 2024) and the data dictionary was obtained from the National Snow and Ice Data Center (NSIDC; https://nsidc.org/data/icesat-2/documents, accessed on 3 September 2024).

### 2.4. Copernicus DEM

The Copernicus DEM is a DSM produced from radar satellite data acquired by the TanDEM-X mission between December 2010 and January 2015. The Copernicus DEM GLO-30 product covers the entire global landmass with a resolution of 30 m and a vertical datum of the Earth Gravitational Model 2008 (EGM2008). The quality of Copernicus DEM GLO-30 is superior to the SRTM with the same resolution [34]. In our study, Copernicus DEM was used to provide elevation values for ground control points (GCPs) and for quality control of ICESat-2 LiDAR point cloud. The Copernicus DEM GLO-30 product was obtained from the European Space Agency (ESA; https://spacedata.copernicus.eu/en/web/guest/collections/copernicus-digital-elevation-model, accessed on 3 September 2024).

## 3. Methodology

### 3.1. Overview

The workflow of the building height extraction is shown in Figure 2. First, we used GF-7 forward and backward images to generate GF-7 DSM. After preprocessing and pansharpening the GF-7 multispectral image, building footprints were extracted by Mask R-CNN instance segmentation and RandomForest classification. The GF-7 DSM and the building mask derived from the building footprints were then fed into the DELaMa deep learning model to generate the GF-7 DTM. We modified the original Large Mask Inpainting with Fourier Convolutions (LaMa) model to quantitatively process DSM data. The next step was to calculate the difference between the GF-7 DSM and the DTM as the nDSM. We employed the percentile value of nDSM in building footprints as the result of building height extraction. Finally, the ICESat-2 LiDAR point cloud data was used to validate the extracted building heights.

### 3.2. Building Footprints Extraction

A geometric correction was applied to the GF-7 MUX multispectral image based on the DSM generated from the GF-7 BWD FWD stereo image and the GCPs from the reference basemap. The radiometric calibration and spectral response parameters provided by CRESDA were used to implement radiometric correction according to the relationship L=Gain·DN+Offset, where L is the radiance in $W\cdot m^{-2}\cdot sr^{-1}\cdot \mu m^{-1}$, Gain is the gain coefficient, Offset is the bias coefficient, and DN is the observed value from the satellite sensor. The radiance was then converted to surface reflectance through the FLAASH atmospheric correction. Pansharpening was conducted using the Gram-Schmidt algorithm to fuse the multispectral image with the panchromatic image to increase spatial detail. The preprocessed image was used for subsequent segmentation and classification.

The Mask R-CNN instance segmentation is implemented based on the Faster R-CNN and performs well in building footprint extraction [35,36,37]. The Faster R-CNN outputs the bounding box and class label of each object, while Mask R-CNN replaces Faster R-CNN’s ROI Pooling with ROI Align and adds two convolution layers to generate the mask of each object. In this study, we use Mask R-CNN to extract building footprints. We manually labeled 1704 buildings as training samples and performed data augmentation by randomly changing rotation, brightness, contrast, zoom, and crop. The backbone used was ResNet-50, and the optimal learning rate was extracted from the learning curve during the training process. 10% of the samples were used for validation, and the training was stopped when the model’s validation loss stopped improving. Vegetation and water bodies have unique spectral characteristics. We used the multispectral image and the RandomForest algorithm [38] to extract the evergreen forest, deciduous forest, shrubs, and water bodies in the study area. A total of 943 training samples were manually labeled. The number of trees was set to 200, and the maximum tree depth was set to 100. The obtained vegetation and water polygons were used to eliminate errors in the building footprints extracted by Mask R-CNN. Finally, the building footprints were manually checked and modified to ensure that the accuracy was greater than 90%.

### 3.3. GF-7 Building Height Extraction

#### 3.3.1. GF-7 DSM Generation

We selected 10 GCPs on flat, bare ground throughout the study area, referencing ESRI’s online World Imagery and Google Earth. Elevation values of the GCPs were derived from the Copernicus DEM. The stereo model was calculated using RPC files of GF-7 images, GCPs, and tie points. The SGM was employed in dense matching to generate high-quality geocoded DSM [39]. The vertical datum of the output DSM was converted to EGM2008, while the resolution was set to 0.5 m to avoid loss of accuracy.

#### 3.3.2. DELaMa

We propose the deep learning model DELaMa for DSM editing. This is achieved by embedding color encoding and decoding modules in the LaMa deep learning architecture [40]. The original LaMa model was trained on a geocoding-free 8-bit RGB image set. It encountered difficulties in handling DSM data with high quantization bits. In our study area, for example, the maximum elevation difference is 661 m. When converted to an 8-bit integer, the sampling interval for the elevation values will exceed 2.5 m.

In this study, we designed a two-stage color coding module (Figure 3a). In the first stage, the DSM of the built-up area was tiled into 500 × 500 m blocks and processed individually to reduce the maximum sampling interval of the elevation values to 0.6 m. In the second stage, each DSM block was color-coded in accordance with the elevation values, ensuring precise correspondence with the 500 colors of the CMRMAP color map. The CMRMAP employs a combination of different colors to achieve a continuous color transition while maintaining the monotonicity of the lightness [41,42]. Its RGB components are shown in Figure 3b. Ultimately, the maximum sampling interval of the elevation values was reduced to 0.3 m.

Clusters of low-rise buildings in rural areas can result in fairly large building masks, necessitating a wide receptive field of deep learning networks. In conventional, fully convolutional models, the effective receptive field grows slowly, and many layers in the network lack global context, especially in the early layers. The large mask inpainting (LaMa) architecture uses fast Fourier convolutions (FFC) to achieve a receptive field over the entire input image. It uses high receptive field loss functions to promote the consistency of global structures. LaMa was trained with large masks and can be applied to images with higher resolution than those used in training [40]. The LaMa model was trained on a subset of 4.5 M images from the Places-Challenge dataset using PyTorch, PyTorchLightning, and Hydra.

In Figure 4, m denotes building masks obtained by expanding building footprints by 10 m to ensure complete coverage. x denotes the color-encoded DSM, $x^{\prime }=stack(x⨀m,m)$ denotes a four-channel tensor consisting of the masked image and the mask. The input $x^{\prime }$ is processed by the feed-forward inpainting network $f_{\theta }$ to yield the three-channel color image $\hat{x}$. In FFC, the channels are split into two parallel branches. The local branch uses conventional convolutions, whereas the global branch uses real FFTs to account for the global context. The steps are as follows:Applies real FFT2d to the input tensor and concatenates real and imaginary parts:
(1)$$Real\;FFT2d:R^{H\times W\times C}\to C^{H\times \frac{W}{2}\times C}$$
(2)$$ComplexToReal:C^{H\times \frac{W}{2}\times C}\to R^{H\times \frac{W}{2}\times 2C}$$

2.Convolution in the frequency domain:


(3)
$$ReLU\circ BN\circ Conv1\times 1:R^{H\times \frac{W}{2}\times 2C}\to R^{H\times \frac{W}{2}\times 2C}$$


3.Inverse transform to recover spatial structure:


(4)
$$RealToComplex:R^{H\times \frac{W}{2}\times 2C}\to C^{H\times \frac{W}{2}\times C}$$



(5)
$$Inverse\;Real\;FFT2d:C^{H\times \frac{W}{2}\times C}\to R^{H\times W\times C}$$


Finally, the results of the local and global branches are merged to generate the final output. The loss function $L_{final}$ is the weighted sum of four parts:(6)$$L_{final}=\kappa L_{Adv}+\alpha L_{HRFPL}+\beta L_{DiscPL}+\gamma R_{1}$$
where $L_{Adv}$ is the adversarial loss, and $L_{DiscPL}$ is the discriminator-based perceptual loss; both are responsible for generating natural local details. $L_{HRFPL}$ is the high receptive field perceptual loss, which is responsible for the supervised signal and consistency of the global structure, and $R_{1}$ is the gradient penalty.

The color of some pixels on the processed image $\hat{x}$ deviated from the CMRMAP (Figure 3c). We projected them to the nearest position on the CMRMAP curve based on the distance in RGB space:(7)$${Dist}_{RGB}=\sqrt{{\left(R_{\hat{x}}-R_{CMRMAP}\right)}^{2}+{\left(G_{\hat{x}}-G_{CMRMAP}\right)}^{2}+{\left(B_{\hat{x}}-B_{CMRMAP}\right)}^{2}}$$
where $R_{\hat{x}}$, $G_{\hat{x}}$, and $B_{\hat{x}}$ represent the values of $\hat{x}$ in the red, green, and blue channels, respectively. $R_{CMRMAP}$, $G_{CMRMAP}$, and $B_{CMRMAP}$ represent the values of CMRMAP in the red, green, and blue channels, respectively. Finally, using the decoding module, the images were converted and mosaicked into a geocoded 32-bit floating point DTM through the reverse process of the encoding module. The model was trained on eight NVIDIA^®^ V100 GPUs for about 240 h. The rest of the experiment was carried out on a computer equipped with an NVIDIA^®^ GeForce RTX 3090 Ti GPU and 64 GB of memory.

#### 3.3.3. Extraction of Building Height from nDSM

The nDSM was obtained by calculating the difference between the DSM and the DTM. In the DSM, some values were generated by interpolating from neighboring values because stereo-matching failed at these locations. In addition, a large number of buildings in the study area have pitched roofs, but elevation data from stereo matching may not be available at the ridges. To better represent the building heights, we ignored the nDSM data at locations where stereo matching failed and then sorted the nDSM values in each building footprint, using the height of the 90th percentile as the building height.

### 3.4. ICESat-2 Building Height Extraction

The ICESat-2 ATL03 product provides the longitude, latitude, elevation, and timestamp of each photon. The ATL08 product is based on the ATL03 and uses the DRAGANN algorithm to classify photons as noise, ground, canopy, and top of canopy. We linked ATL03 and ATL08 based on the index field to get the elevation, position, and classification flag of each photon [43] and then converted the photon’s elevation datum to EGM2008. Only data from the strong beams were used, which accounted for 61% of all ICESat-2 data in the study area. Photons classified as noise in ATL08 and photons with an elevation difference greater than 50 m or less than −10 m from the Copernicus DEM were removed. Consistent with the nDSM treatment, we used the 90th percentile photon elevation in the building footprint as the roof elevation of the building and the 10th percentile photon elevation in the 10 m buffer zone around the building as the ground elevation. ICESat-2 building heights were obtained by calculating the difference between roof and ground elevations [13,44,45].

### 3.5. Evaluation Indicators

The least-squares regression was used to assess the correlation between the extracted building heights and the reference building heights. The accuracy of the building heights was evaluated by the mean absolute error (MAE) and the root mean square error (RMSE):(8)$$MAE=\frac{1}{N}\sum_{i=1}^{N}\left|{\hat{h}}_{i}-h_{i}\right|$$
(9)$$RMSE=\sqrt{\frac{1}{N}\sum_{i=1}^{N}{\left({\hat{h}}_{i}-h_{i}\right)}^{2}}$$
where *N* denotes the number of samples, ${\hat{h}}_{i}$ denotes the height of the $i_{th}$ building, and $h_{i}$ denotes the reference height.

## 4. Results

### 4.1. Building Footprints of the Study Area

After filtering by the RandomForest classification results, an extensive manual inspection was carried out to remove remaining patches on roads or bare ground. Finally, a total of 50,925 building footprints were obtained, as shown in Figure 1a. The mean area of the building footprints is 154.52 m^2^, and the median area is 75.75 m^2^. A factory has the largest area of about 25,000 m^2^. The total area of the building footprints is about 7.87 km^2^. In the valleys, buildings are spread out at the foot of the hills, while in the relatively flat areas, they are clustered and tend to be close to the road.

### 4.2. Performance of DELaMa

Figure 5 shows comparisons of DELaMa with three other commonly used DSM editing methods using the same mask (Figure 5d) as the processing area. To show details, they are presented as multidirectional hillshade. Figure 5a shows the original GF-7 DSM. Figure 5b shows the result filled in using other DEM sources. Here, we used the Copernicus DEM, which was interpolated to a resolution of 0.5 m. The result reflects some topographic relief information. It has a smooth surface and obvious errors at the edges. Figure 5c shows the result of the plane fitting. The entire processed area is flat and smooth. Figure 5e shows the result of the terrain filter, which smoothed out the protrusions of buildings while preserving certain terrain details. Figure 5f shows the result of DELaMa. Compared with the other three methods, our method not only preserved the most topographic details but also generated a reasonable prediction of the topographic details inside the mask based on the topographic features of the entire area.

### 4.3. Results of Building Height Extraction

The building height map of the entire study area is shown in Figure 6. The result is aggregated to a 50 m ground sampling distance (GSD) for visualization. In general, the taller buildings are mainly found in the flat, open area to the south of the study area, which is a town center with many workshops and multistory buildings. In the southwestern valley, there are some buildings over 15 m high, which are large production facilities around a mine. In another town center to the north, several school buildings and workshops are over 15 m high. Most buildings in remote valleys are between 3 and 5 m. The shortest buildings are the greenhouses, but some of the greenhouses are underestimated, as they are close to 0 m. Figure 7 shows examples of building height extraction. The first column is the GF-7 multispectral image, the second column is the GF-7 DSM, the third column is the nDSM, and the fourth column is the extracted building height. It can be seen that the larger buildings are usually taller, the bungalows are of a similar height, and the outbuildings are lower.

## 5. Discussion

### 5.1. Building Height Accuracy Assessments

The buildings whose footprints intersect the ICESat-2 beams were employed as the validation buildings. To ensure the reliability of the validation, the following samples were excluded from the validation set: (1) the number of ICESat-2 photons or the nDSM values from successful stereo matching in the building footprint of less than 10 or located only near the edges of the building; (2) buildings obscured by trees; (3) closely surrounded by other buildings, with no ground-reflected ICESat-2 photons in the buffer; and (4) buildings less than 1.5 m in height as measured by ICESat-2. Finally, a total of 383 buildings were used for validation (Figure 1a). The scatter plot of the building heights from GF-7 against the building heights from ICESat-2 is shown in Figure 8, with an R^2^ of 0.83, an MAE of 1.81 m, and an RMSE of 2.13 m. Figure 9 and Appendix A show the quality control and inspection of the validation process, demonstrating the reliability and robustness of the ICESat-2 building height extraction method. The ICESat-2 photons are shown as green dots. The 90th percentile values of photon elevation in building footprints and the 10th percentile values of photon elevation in buffers are marked with red and blue dashed lines, respectively.

### 5.2. Impact of DSM Editing Method

Two zones with rugged terrain were selected to demonstrate the impact of DSM editing method on building height extraction. The buildings used for comparison are bungalows at the foot of the hill, representing a common scenario where building height extraction is affected by terrain. The selected bungalows were scanned by the laser beam of ICESat-2, as shown in Figure 10a. Figure 10b shows the GF-7 DSMs, and Figure 10c–f shows the DTMs generated by different DSM editing methods, including the Copernicus DEM filling, plane fitting, terrain filter, and DELaMa, respectively. The DELaMa model was shown to produce the most refined DTM, followed by the terrain filter method. We extracted building heights for both zones based on the DTMs generated by the three reference methods, using the same processing flow as in the methodology section, and compared them with the reference building heights extracted from the ICESat-2 photon point cloud, as shown in Table 1. In both zones, the deviations of the building heights extracted based on DELaMa are the smallest, and the deviations of the results based on the terrain filter are slightly larger, consistent with the comparisons in Figure 10. The Copernicus DEM filling and plane fitting methods both deviate by more than 1 m. In Zone 1, the elevation from the Copernicus DEM filling exceeds the DSM, resulting in a negative building height. However, in scenarios with larger filling areas, such as Zone 2, the Copernicus DEM filling produces a better result than the plane fitting, attributed to its elevation information.

## 6. Conclusions

In this study, we extracted building heights in rural areas using images from China’s GF-7 stereo mapping satellite. The Mask R-CNN was used to extract building footprints, and the DELaMa model was proposed to process the DSM. The extracted building heights were validated using ICESat-2 LiDAR point cloud data. The experiment and quantitative validation were carried out in the mountainous area of Pingquan in northern China, and the validation resulted in an R^2^ of 0.83, an MAE of 1.81 m, and an RMSE of 2.13 m. The study of large-scale building height extraction in rural areas can help to promote the use of building height information extracted from satellite data in disaster management. Considering the potential for global coverage of GF-7 imagery and the effectiveness of DELaMa, our future work will focus on extracting building height in rural areas with different architectural characteristics, for example, in southern China and central Asia. Finer processing, such as morphological filtering and 3D segmentation of the GF-7 DSM data, would be beneficial in humid climates where dense vegetation tends to obscure buildings. Deep learning methods could also be employed to address missing values resulting from occlusion by vegetation, terrain, or other structures. We will also test and compare the performance of the proposed method on other high-resolution optical stereo images, such as WorldView, and further improve the robustness of the method.

## Figures and Tables

**Figure 1 sensors-24-06076-f001:**
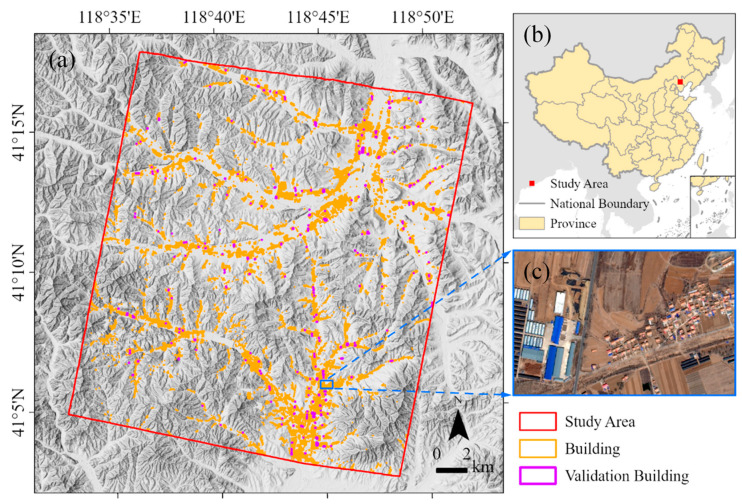
The study area: (**a**) extent of the study area and building footprints, overlay on the Copernicus DEM hillshade layer; (**b**) location of the study area; and (**c**) demonstration of building types in the study area.

**Figure 2 sensors-24-06076-f002:**
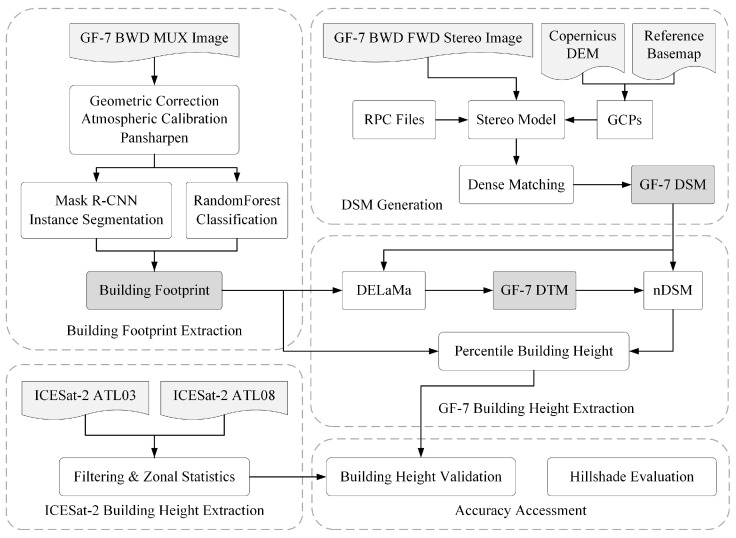
Workflow of the building height extraction. Important intermediate products are marked with a darker background.

**Figure 3 sensors-24-06076-f003:**
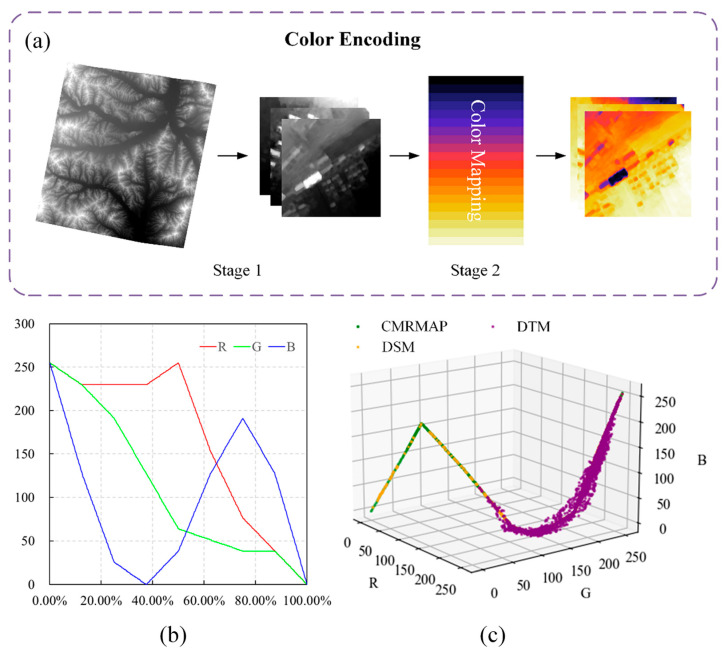
The color encoding module: (**a**) two-stage color encoding; (**b**) RGB components of CMRMAP; (**c**) CMRMAP, digital surface model (DSM), and digital terrain model (DTM) in RGB color space.

**Figure 4 sensors-24-06076-f004:**
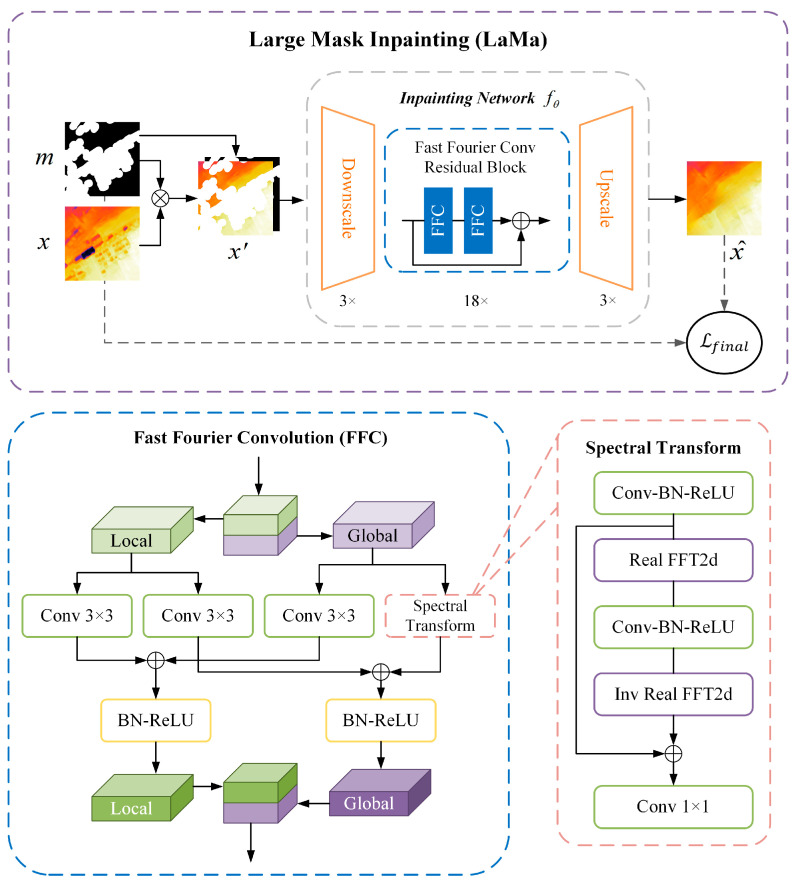
The architecture of Large Mask Inpainting (LaMa) [40].

**Figure 5 sensors-24-06076-f005:**
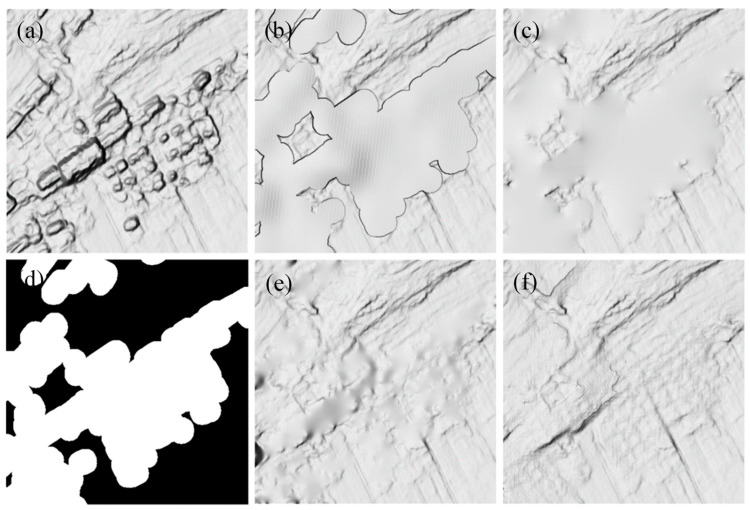
Comparisons of different DSM editing methods. The results are presented as multidirectional hillshade to show details: (**a**) original GF-7 DSM; (**d**) building mask, derived from the building footprints; (**b**) filled in using Copernicus DEM; (**c**) plane fitting; (**e**) terrain filter; and (**f**) our method, DELaMa.

**Figure 6 sensors-24-06076-f006:**
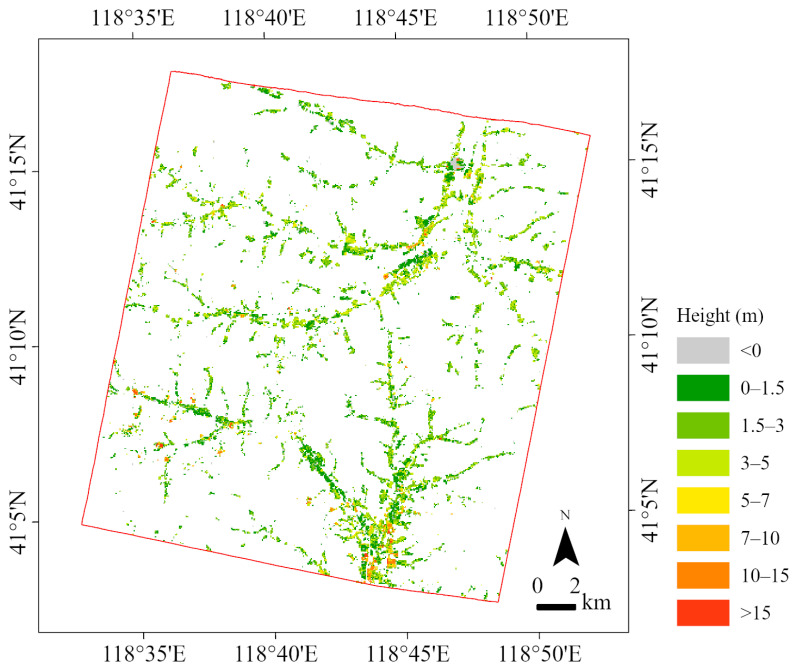
The building height map of the study area. Aggregated to 50 m ground sampling distance (GSD) for visualization.

**Figure 7 sensors-24-06076-f007:**
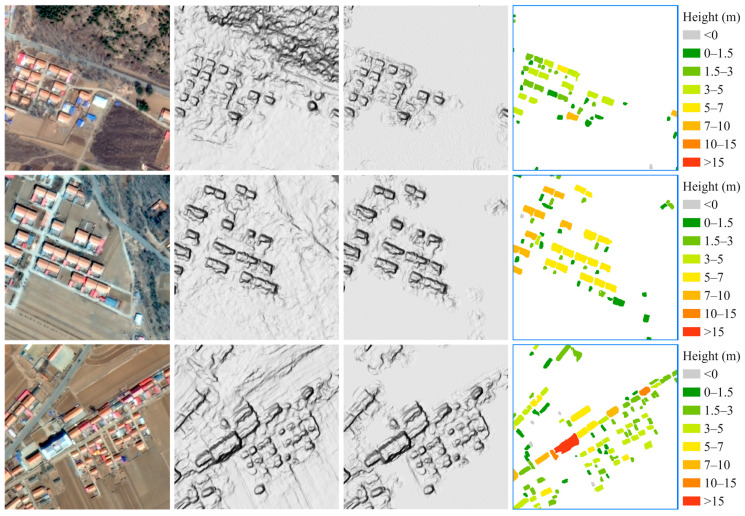
Examples of building height extraction. The first column is the GF-7 multispectral image, the second column is the GF-7 DSM, the third column is the normalized digital surface model (nDSM), and the fourth column is the extracted building height.

**Figure 8 sensors-24-06076-f008:**
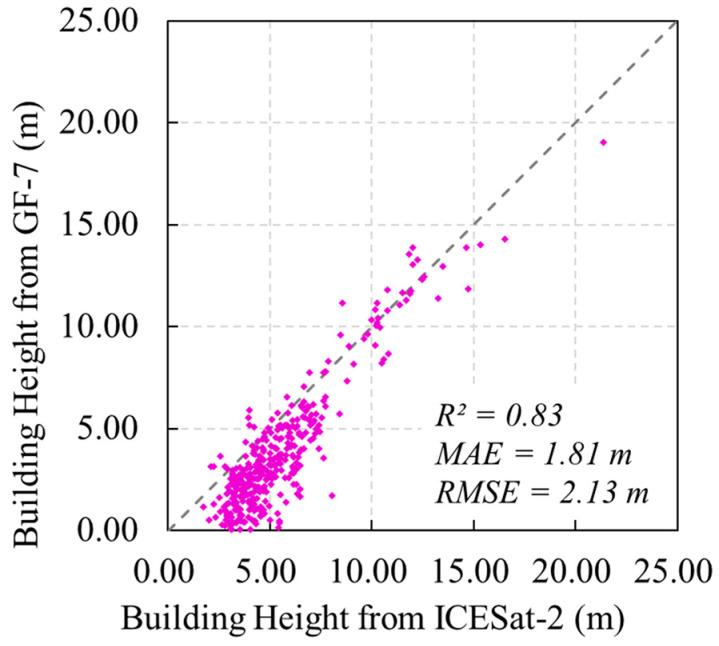
Scatter plot of the building heights from GF-7 against the building heights from ICESat-2.

**Figure 9 sensors-24-06076-f009:**
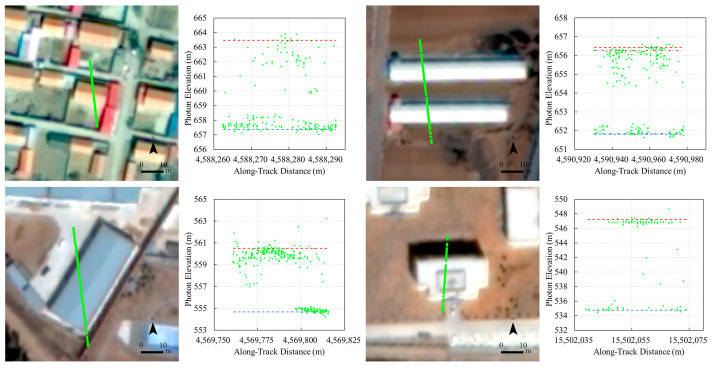
Quality control and inspection of the validation process.

**Figure 10 sensors-24-06076-f010:**
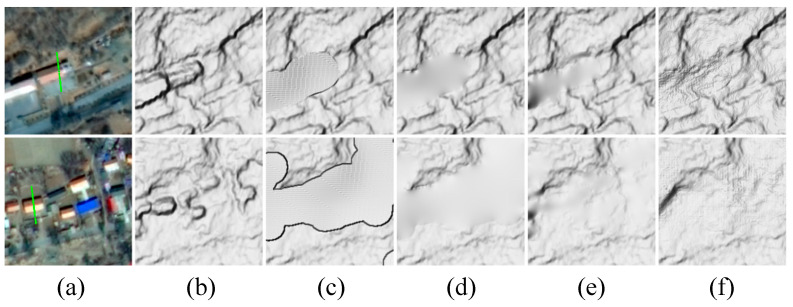
Demonstrations of the impact of DSM editing methods on building height extraction in rugged terrain. The topographic data are presented as multidirectional hillshade to show details: (**a**) images of Zone 1 and Zone 2; (**b**) GF-7 DSM; (**c**) Copernicus DEM filling; (**d**) plane fitting; (**e**) terrain filter; and (**f**) DELaMa.

**Table 1 sensors-24-06076-t001:** Building heights generated based on different DSM editing methods (in meters).

	Reference	DEM Filling	Plane Fitting	Terrain Filter	DELaMa
Zone 1	5.36	−0.18	3.10	4.88	4.94
Zone 2	4.81	3.59	3.29	3.84	4.44

## Data Availability

Data are available on request due to restrictions.

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
