# Peer review of "A Method for Extracting High-Resolution Building Height Information in Rural Areas Using GF-7 Data"

_sensors, 2024, doi:10.3390/s24186076_

Round 1

Reviewer 1 Report

Comments and Suggestions for Authors

Considering the scarcity of research and the low accuracy in estimating rural building heights using GF-7 stereo images, this study developed a reliable method for extracting building heights based on GF-7 stereo imagery. In particular, the DELaMa model was proposed to optimize the DEM and improve nDSM estimation accuracy, addressing the complexity of rural terrain. However, the manuscript still has some shortcomings:

1. Introduction of the study area: Compared to previous studies on extracting building heights based on GF-7 stereo images, this paper focuses on rural areas, highlighting the greater complexity of rural terrain compared to urban terrain. However, the study area introduction does not emphasize its representativeness, and it is recommended to supplement this.

2. Data introduction: It is recommended to include the sources of different datasets and related preprocessing steps.

3. Accuracy validation: The paper estimates building heights based on ICESat-2/ATL08 height products by extracting building and ground photons using building footprint data and a specific buffer zone. However, it is unclear whether other photons (e.g., vegetation photons) that could cause height estimation bias have been excluded. Therefore, I believe it is necessary to first validate the accuracy of building height extraction based on ICESat-2/ATL08 to ensure its reliability before using it as validation data for this study.

4. Results: As building footprint data is an important input parameter for DELaMa, it is recommended to include relevant descriptions of its results in the results section.

5. Discussion: The DEM produced by different methods has a significant impact on the accuracy of building height extraction, and the DELaMa model proposed in this study is designed to handle complex terrain (rural areas). Therefore, the discussion should focus on analyzing the differences in building height estimation accuracy under DEMs generated by different methods. Analyzing the extraction accuracy of different building types is not meaningful, as the described workshops and multistory buildings, which show better estimation accuracy, are also common in urban areas.

Reviewer 2 Report

Comments and Suggestions for Authors

This article presents a novel approach to extracting building heights in rural areas using satellite imagery from China's Gaofen-7 (GF-7) satellite. Building height is crucial for various applications, including disaster management, damage assessment, population modeling, and urbanization studies.

This method holds promise for enhancing disaster management and urban planning efforts in less densely populated areas.

Suggestions:

- Rural areas often have significant vegetation and other features that might occlude buildings. Addressing the impact of occlusions could increase accuracy.

- Explaining how the model handles edge cases, such as buildings partially obscured by other structures or those on steep terrain, could further refine the method.

- While the focus is on GF-7 imagery, a comparative analysis with data from other satellites, such as Sentinel or WorldView, could highlight the strengths and limitations of using GF-7 and potentially improve the robustness of the method.

- The study could provide more details on the computational resources required for the method.
